# Gene Networks Involved in Plant Heat Stress Response and Tolerance

**DOI:** 10.3390/ijms231911970

**Published:** 2022-10-09

**Authors:** Ling-Zhi Huang, Mei Zhou, Yan-Fei Ding, Cheng Zhu

**Affiliations:** Key Laboratory of Specialty Agri-Product Quality and Hazard Controlling Technology of Zhejiang Province, College of Life Sciences, China Jiliang University, Hangzhou 310018, China

**Keywords:** heat stress, plant, response, gene expression, regulation

## Abstract

Global warming is an environmental problem that cannot be ignored. High temperatures seriously affect the normal growth and development of plants, and threaten the development of agriculture and the distribution and survival of species at risk. Plants have evolved complex but efficient mechanisms for sensing and responding to high temperatures, which involve the activation of numerous functional proteins, regulatory proteins, and non-coding RNAs. These mechanisms consist of large regulatory networks that regulate protein and RNA structure and stability, induce Ca^2+^ and hormone signal transduction, mediate sucrose and water transport, activate antioxidant defense, and maintain other normal metabolic pathways. This article reviews recent research results on the molecular mechanisms of plant response to high temperatures, highlighting future directions or strategies for promoting plant heat tolerance, thereby helping to identify the regulatory mechanisms of heat stress responses in plants.

## 1. Introduction

The ambient air temperature caused by global warming continues to rise, which makes high-temperature stress ever more serious. High-temperature stress not only restricts the growth and development of plants, but also reduces the yield of crops, threatens food security, and endangers human life. Each plant has its own optimum temperature for each stage of growth and development. The optimum temperature for rice seed germination, for example, is 28–32 °C, whereas the optimum temperature at the heading stage is 25–35 °C. Stress caused by supra-optimal temperature leads to the loss of yield and grain quality [1,2]. Plants have evolved a number of cellular and physiological mechanisms by which to cope with high-temperature stress. The cell membrane system is central to thermal damage and resistance. High-temperature stress damages the cell membrane lipids of plants, causes an increase in membrane permeability, and affects the thermal conductivity of the membrane stability and osmoregulation, eventually leading to cell death [3,4]. Studies have pointed out that high-temperature stress significantly inhibits the photosynthetic rate and respiration [5]. High temperatures disrupt the balance between the production and scavenging of reactive oxygen species (ROS) in cells, resulting in the accumulation of oxides such as H_2_O_2_ and malondialdehyde (MDA), exposing plants to oxidative damage during oxidative stress [6,7].

So far, a large amount of published research has described the molecular mechanisms of plant response to high-temperature stress (Figure 1, Table 1 and Table 2). Many heat stress-induced genes have been identified based on microarray and RNA-sequencing analyses in *Arabidopsis* (*Arabidopsis thaliala*) and rice (*Oryza sativa*), and among various plant species [8,9]. Their gene products can be divided into three groups. The first group includes proteins that most probably function in heat stress tolerance, including ubiquitin ligases, RNA helicases, heat shock proteins (Hsps), ion transporters, and aquaporins (AQPs) [10,11,12,13]. Heat stress causes metabolic alterations affecting protein and RNA stability. Hsps, ubiquitin ligases, and RNA helicase play important roles in protein and RNA metabolism under heat stress. Heat stress affects the expression of membrane channel proteins (such as AQPs and SUTs) involved in the transport of water, small solutes, and carbohydrates. The second group is comprised of regulatory proteins, such as receptor-like protein kinases (RLKs), mitogen-activated protein kinases, calcium-dependent protein kinases (CDPKs), transcription factors, and so on [7,14,15]. They play important roles in the regulation of signal transduction, such as heat-responsive gene expression under conditions of heat stress. In addition, heat stress alters membrane fluidity which activates Ca^2+^ channels, resulting in an influx of Ca^2+^ [16]. Ca^2+^ signals are transduced by Ca^2+^ sensors to different substrates, activating signal transduction pathways in plants in response to heat stress [7,15]. Under heat stress, low concentrations of hormones as a signal molecule can regulate various physiological and biochemical reactions of plants [17]. The third group includes products such as non-coding RNAs (ncRNAs) that can also regulate heat stress response, including microRNAs (miRNAs) and long non-coding RNAs (lncRNAs) [18,19]. MiRNAs and lncRNAs can interact with other molecules (DNA, RNA, and proteins) to achieve the internal regulatory lncRNA-miRNA-mRNA network and hence improve the heat tolerance of plants [20,21].

As a consequence, these gene products form regulatory networks that regulate gene transcription and translation, activate antioxidant defenses, induce signaling pathways, and maintain near-normal metabolism under stressful circumstances [53]. In this review, we summarize the information acquired to date about the molecular mechanisms of plant response to high-temperature stress, thereby helping to identify the detailed regulatory networks that underlie heat stress response in plants. The specific functions of these genes in conferring heat stress tolerance are also discussed, mainly in the model plants, *Arabidopsis* and rice.

## 2. Genes Involved in RNA and Protein Stability

### 2.1. RNA Helicase

RNA helicase plays an important role in RNA metabolism [54]. It is mainly involved in the regulation of RNA structure formation, ribosome formation, and RNA and protein processing. RNA helicase uses the energy released by ATP hydrolysis to transform the RNA, ribonucleoprotein complex to correctly fold RNA and maintain the RNA dynamic balance under abiotic stress [11].

The eukaryotic initiation factor-4A1 (eIF4A1) is a DEAD-box RNA helicase protein, which is known to mediate protein–protein interactions or interactions with ATP or RNA. Mostly, the binding and hydrolysis of ATP play a major role in RNA binding and duplex unwinding [55]. The expression of *OseIF4A1* was significantly upregulated under heat stress in rice seedlings [31]. Strong ATP/Mg^2+^ binding under heat stress suggested a role for OseIF4A1 in heat response. BrRH22 is a chloroplast-targeted DEAD-box RNA helicase in Chinese cabbage (*Brassica rapa*) [34]. The expression of *BrRH22* was greatly affected by high-temperature stress, and a dehydration-responsive element-binding protein 2 (*DREB2*) was significantly upregulated in *BrRH22*-overexpressed *Arabidopsis* under high-temperature stress. In addition, SlDEAD30 and SlDEAD31 are two DEAD-box RNA helicases involved in the abiotic stress response in tomato [46]. *SlDEAD30* was highly expressed in roots and mature leaves, whereas *SlDEAD31* was expressed in all the tissues. Under heat stress, the expression of *SlDEAD30* was downregulated, and that of *SlDEAD31* was upregulated. *SlDEAD31*-overexpressed plants had a higher rate of photosynthesis and maintained a stable water content under heat stress conditions. In response to heat stress, the survival rate of *SlDEAD31*-overexpressed lines was higher than that of the WT, confirming that *SlDEAD31* could improve heat tolerance.

Thermotolerant Growth Required1 (TOGR1) is a DEAD-box RNA helicase that functions as an intrinsic pre-rRNA chaperone in rice, and its expression and activity are enhanced by temperature increases [11]. Under heat stress, misfolding occurs in rRNA, which destroys the interaction between the rRNA precursor and its interacting proteins. At this time, *TOGR1* expression is enhanced in the nucleolus, which helps the rRNA precursor to effectively fold and interact with corresponding proteins to ensure the normal production of rRNA. *TOGR1*-overexpression significantly improved rice growth under heat stress. In order to study the role of *TOGR1* in other crops, Yarra et al. successfully produced transgenic plants of non-heading Chinese cabbage overexpressing the *OsTOGR1* gene [56]. The overexpression of *TOGR1* in Chinese cabbage improved the photosynthesis and inhibited the photooxidation of chlorophyll, so that transgenic Chinese cabbage maintained high chlorophyll content under heat stress, indicating that *TOGR1* positively regulated thermotolerance in plants.

### 2.2. Heat Shock Factors and Heat Shock Proteins

The heat shock factors (Hsfs) and Hsps maintain near-normal cell physiology and metabolism under heat stress conditions and play key roles in achieving stress tolerance in plants. Hsfs can recognize the heat shock element in the upstream promoter region of *Hsp* genes involved in the heat stress response, and upregulate their transcription. Under heat stress, the synthesis of normal proteins in plants decreases, and the signal pathways induce a sharp increase in the concentration of Hsps, helping to fold newly synthesized proteins or to protect existing proteins that may become misfolded during heat stress.

The first plant Hsf was cloned from tomato (*Solanum lycopersicum*), although more Hsfs were later reported from other plant species [57]. According to the characteristics of the HR-A/B region, plant Hsfs can be classified into HsfA, HsfB, and HsfC [58]. HsfAs possess activator motifs or aromatic and hydrophobic amino acid residues close to the NES at the C-terminal region, while HsfBs have a repressor domain. The knowledge of the role of HsfCs is limited. Most of the known roles of Hsfs in plant high-temperature stress are concentrated in Class A Hsfs, and a few in Class B Hsfs and Class C Hsfs. In pepper (*Capsicum annuum* L.), the CaHsfA1d protein is located in the nucleus [40]. The survival rate of *CaHsfA1d*-silenced pepper seedlings in response to heat treatment was lower than that of the control group (*TRV2:00*), indicating that silencing of *CaHsfA1d* reduced the thermotolerance of pepper. After heat treatment, the germination rate and fresh weight of *CaHsfA1d*-overexpressed *Arabidopsis* seeds were higher than those of the corresponding wild type (WT). The overexpression of the *CaHsfA1d* gene enhanced the expression of *AtHsfA2/3*, *AtDREB2A*, and *AtHsps* genes and genes related to glutathione synthesis (*AtGSTU5* and *AtGPX3*) in transgenic *Arabidopsis*. Zhang et al. cloned *ZmHsf01*, a member of the *HsfA2* subclass in maize (*Zea mays* L.) [39]. Under high-temperature treatment, *ZmHsf01* expression located in the nucleus was significantly upregulated in young roots and leaves. *Arabidopsis* seedlings overexpressing *ZmHsf01* had a higher chlorophyll content and survival rate than WT seedlings following heat stress, indicating that *Arabidopsis* overexpressing *ZmHsf01* had greater heat tolerance. The chlorophyll content and survival rate of the *Arabidopsis ZmHsf01/athsfa2* complementary strain, in which the native *Arabidopsis* gene was knocked out but the corresponding maize heat shock factor gene was overexpressed, were greater than in either *athsfa2* or WT. This finding implied that *ZmHsf01* could partially or completely compensate for the heat tolerance defect of the *Arabidopsis athsfa2* mutant. Other reports found that overexpressing *ZmHsf05* could also compensate for the reduced heat tolerance of the *Arabidopsis athsfa2* mutant. Both *ZmHsf05*-overexpressed and *ZmHsf06*-overexpressed *Arabidopsis* plants had greater heat tolerance than the WT plants [59,60]. Tomato (*Lycopersicon peruvianum*) HsfB1 is a novel type of coactivator cooperating with tomato HsfA1. HsfB1 responded to heat stress by maintaining and/or restoring the expression of housekeeping genes (such as *Hsp17.6*) [45].

As an important molecular chaperone, Hsps maintain normal cell physiology and metabolism under stress conditions and play a key role in stress tolerance. OsHSP20 is located in type I cytoplasm in rice cells [12]. Under heat stress, the transcription of *OsHSP20* responded very rapidly, peaking at approximately 4000-fold more transcripts than in the unstressed control within one hour. Under heat stress, overexpression of *OsHSP20* increased the germination rate, plant height, and chlorophyll content of transgenic rice under heat stress conditions, thus improving the heat tolerance during germination and growth. OsHsp18.0 was also located in the cytoplasm, especially at the edge of the nucleus, implying a potential role for OsHsp18.0 in nucleo-cytoplasmic trafficking [29]. Rice lines with overexpressed or silenced *OsHsp18.0*, the latter resulting from RNA interference, were constructed to study the effects on heat tolerance. The overexpressed *OsHsp18.0* rice plants showed greater heat tolerance than the WT, whereas the silenced plants showed a decrease in heat tolerance.

One-year-old grape (*Vitis vinifera*) plants under heat stress for two hours at sunrise and sunset [36]. Short heat stresses (2 h) were applied day and night to vines bearing clusters sequentially ordered according to the developmental stages along their vertical axes. The study found that *HSFA6B* was consistently upregulated under heat stress, but this induction was more pronounced at night than day. Several transcripts coding for members (*VIT_04s0008g06000*, *VIT_18s0001g03120*, *VIT_18s0001g05850*, *VIT_16s0013g00980*, and *VIT_16s0013g01000*) of ethylene family response factors (ERFs) acting upstream of HSFs, were all activated under heat stress. Transcriptome analysis of leaves at the jujube (*Ziziphus jujuba* Mill.) seedling stage revealed that multiple HSPs such as HSP17, HSP18, HSP21, HSP22, HSP23, HSP26, HSP70, HSP83, HSP90, HSF30, and HSC-2 were upregulated [37]. These results suggested that Hsfs and Hsps were crucial in plants under heat stress.

### 2.3. Ubiquitin Ligases

The ubiquitin–proteasome pathway is an important way to regulate protein stability in the signal transduction process under high-temperature stress [61]. Ubiquitin ligases play a specific role in recognizing target protein substrates in the ubiquitin–proteasome pathway, and selectively degrading the key components of stress signals, thus negatively or positively regulating plant response to stresses.

Kim et al. reported that a rice heat-induced RING finger protein 1 (*OsHIRP1*) represented a ring-HC type E3 ligase [10]. Under heat stress, the expression of *OsHIRP1* increased significantly and expression levels of *OsAKR4* and *OsHRK1* were significantly decreased in 14-day-old rice seedlings. An in vitro ubiquitination assay showed that two substrate proteins, Aldo/Keto Reductase 4 (OsARK4) and HIRP1-Regulated Kinase I (OsHRK1), were ubiquitinated by OsHIRP1 E3 ligase, and each was degraded by the Ub/26S proteasome system. *OsHIRP1*-overexpressed *Arabidopsis* exhibited higher seed germination rates and survival rates, compared with WT, under heat stress. The phenotype analysis of *OsHIRP1*-overexpressing *Arabidopsis* showed thermotolerance under heat stress. Expression of some heat stress-inducible genes (*HsfA3*, *Hsp17.3*, *Hsp18.2*, and *Hsp20*) were also upregulated in the *OsHIRP1*-overexpressing *Arabidopsis* under heat stress (Figure 2).

Kim et al. reported another rice protein (drought-, heat-, and salt-induced RING finger protein 1 (OsDHSRP1)) acting as a ring-H2 E3 ligase [30]. The transcription level of *OsDHSRP1* was significantly upregulated in heat-stressed 14-day-old rice seedlings. OsDHSRP1 E3 ligase ubiquitinated two interacting substrates in rice, glyoxalase (OsGLYI-11.2) and abiotic stress-induced cysteine proteinase 1 (OsACP1), and each was degraded by the Ub/26S proteasome system. *Arabidopsis* plants overexpressing *OsDHSRP1* showed increased sensitivity to heat stress, and their germination percentage and root length under heat stress were lower than those of control plants. This was because the degradation of the OsGLYI-11.2 protein maintained a lower acetaldehyde level, thus increasing methylglyoxal and ROS content in the transgenic *Arabidopsis* plants (Figure 2).

A protein with the RING domain and Tmemb_185A domain (abbreviated as AtPPRT1) is a putative C3HC4 zinc-finger ubiquitin E3 ligase. It positively regulated the expression of several heat stress-inducible genes (*AtHSP21*, *AtHSFA7a*, and *AtZAT12*) under heat stress, and enhanced heat tolerance by reducing the accumulation of ROS in *Arabidopsis* [62]. AtPUB48 is an E3 ubiquitin ligase with a U-box and Armadillo-repeats [22]. *AtPUB48*-overexpressed *Arabidopsis* plants exhibited increased basal and acquired thermotolerance in terms of seed germination and seedling growth. In addition, wheat (*Triticum aestivum*) F-Box Protein Gene (*TaFBA1*) encodes a subunit of the Skp1-Cullin-F-box E3 ligase complex, which interacted with *Triticum aestivum* stress-responsive protein 1 (TaASRP1) and other proteins [43]. *TaASRP1*-overexpression in wheat improved the enzymatic antioxidant system and reduced cell damage under heat stress. In *Sorghum* (*Sorghum bicolor*), the heat- and cold-induced RING finger protein 1 (SbHCI1) possesses E3 ligase activity and can interact with and ubiquitinate the substrates (b14-3-3, SbbHLH065, and SbBGLU1) [42]. Sb14-3-3 is a protein located in the cytoplasm, whereas SbbHLH065 and SbBGLU1 are located in the nucleus. In response to heat treatment, the survival rate of *SbHCI1*-overexpressed plants was significantly higher than in the corresponding WT plants, indicating that *SbHCI1* positively regulated heat stress tolerance. The rice soluble ubiquitin-specific protease (OsUBP21) negatively regulated the heat shock response under heat stress [63]. Its homologous gene *AtUBP13* in *Arabidopsis* also exhibited protein deubiquitination activity and negatively regulated heat shock response. Knocking down the expression of *OsUBP21* in rice or knocking out *AtUPB13* in *Arabidopsis* by T-DNA insertion enhanced the tolerance of the mutants in response to heat stress, compared with the WT.

## 3. Genes Involved in Substance Transport

Plants coordinate growth and development with nutrient availability. Changes in the external environment affect the synthesis, degradation, and transportation of substances in plants. In recent years, genes involved in substance transport have been isolated and have been shown to play significant roles in responses to abiotic and biotic stresses.

### 3.1. Water Transport

AQPs are membrane channel proteins transporting water and small solutes. Based on the protein sequence homology and membrane localization, plant AQPs are divided into five sub-families: plasma membrane intrinsic proteins (PIPs), tonoplast membrane intrinsic proteins (TIPs), NOD26-like membrane intrinsic proteins, small basic membrane intrinsic proteins, and GlpF-like membrane intrinsic proteins [64]. Recently, it has been reported that AQPs are involved in response to heat stress. In wheat, TaTIPs respond to combined heat and drought stresses, based on the representation of expressed sequence tags in wheat grain-related cDNA libraries [65]. In soybean (*Glycine max*) seedling roots, expression of the aquaporin gene *GmTIP2;6* was upregulated under heat stress [13]. GUS activity, driven by the *GmTIP2;6* promoter, was strongly induced in the heat-treated transgenic *Arabidopsis* plants and accumulated in hypocotyls, vascular bundles, and leaf hairs. While six-month-old strawberries (*Fragaria x ananassa* cv. ‘Camarosa’) pretreated with sodium hydrosulfide were exposed to heat stress for four hours, the expression of the aquaporin gene *FvPIP* was found to be upregulated, allowing the plants to achieve heat tolerance [64]. Under high-temperature stress, the expression of three *AQP* genes (namely *pip2-1*, *pip1-2*, and *tip21*) in leaves of the medicinal plant *Rhazya stricta* was upregulated. These proteins promoted the diffusion of water across the cell membrane, balanced the water within the cell, and improved the water utilization rate, which enhanced the heat tolerance of *R. stricta* [41]. These results contributed new insights into the regulatory mechanisms of thermotolerance in plants.

### 3.2. Photosynthesis and Sucrose Transport

In the process of higher plants’ photosynthesis, chlorophyll biosynthesis, photochemical reaction, electron transfer, and CO_2_ assimilation play important roles in plant heat tolerance. It is well known that photosynthesis is highly sensitive to heat stress. Chlorophyll is the main pigment for photosynthesis light absorbance. PSII is one of the most thermosensitive components of photosynthetic apparatus. Heat stress changes the redox balance of photosynthetic electron transport reactions [66]. Under heat stress conditions, the heat tolerance of maize genotype “*DKC7221*” is based on its higher photosynthetic activity [5]. In the leaves of *DKC7221* seedlings, heat stress did not affect the total chlorophyll content, and the electron transport rate into the plastoquinone pool. The study also found that under heat stress, *DKC7221* was able to maintain a relatively high level of Fv/Fm ratio. The study indicated that the electron transport reaction on the PSII unit remained almost intact under heat stress conditions. A study demonstrated that chloroplast signal recognition particle 43 (cpSRP43) effectively protected several tetrapyrrole biosynthesis proteins (GlutR, CHLH, and GUN4) from heat-induced aggregation and enhanced their stability during leaf greening and heat shock [67]. *MdATG18a* is the autophagy-related gene in apples (*Malus domestica*) [33]. *MdATG18a* transcript was significantly upregulated under heat stress. A study found that *MdATG18a* improved thermotolerance by enhancing autophagic activity and maintaining high levels of photosynthesis. The overexpression of *MdATG18a* in apples enhanced photosynthetic capacity, as shown by the electron transport rates in PSI and PSII, the maximum photochemical efficiency of PSII, and the rate of CO_2_ assimilation. In addition, a tomato chloroplast-targeted DnaJ protein (SlCDJ2) was found to be uniformly distributed in the thylakoids and stroma of the chloroplasts [68]. Within 24 h at 42 °C, the expression of *SlCDJ2* gradually increased in six-week-old tomatoes. SlCDJ2 together with Hsp70 may help protect ribulose-1,5-bisphosphate carboxylase/oxygenase (Rubisco) activity from heat stress. This contributed to maintaining CO_2_ assimilation capacity and enhancing heat tolerance.

Furthermore, sucrose transport also plays an important role in plant heat tolerance. When rice plants are exposed to high temperatures during the grain ripening period, the sink–source balance of carbohydrates is disrupted. The carbohydrates cannot be supplied to the kernels through current photosynthesis after heading, thus affecting rice yield. Recently, several laboratories reported that the level of high-temperature tolerance of rice was related to carbohydrate concentration at the full-heading stage [69,70]. The rice sucrose transport gene (*OsSUT1*) encodes a sucrose transporter, which plays an important role in maintaining the supply of photoassimilates to the filling grains. *OsSUT1* is highly expressed in leaf sheaths, stems and grains after heading, but at only a low level in the roots [71,72]. The expression level of *OsSUT1* in grains between 8 and 30 days after flowering was reduced under high-temperature conditions, suggesting that *OsSUT1* was involved in the efficient maturation of rice [48]. Furthermore, Miyazaki et al. studied the effects of heat stress on the grain quality of heat-tolerant (‘*Genkitsukushi*’) and heat-sensitive rice cultivars (‘*Tsukushiroman*’) during maturation [69]. The non-structural carbohydrate content in the stems (photosynthetic reserves accumulated during the less stressful part of the growing season) of ‘*Genkitsukushi*’ at early maturation was significantly higher than in ‘*Tsukushiroman*’, but greatly decreased under high temperatures. The expression of *OsSUT1* in ‘*Genkitsukushi*’ grain was significantly higher than that of ‘*Tsukushiroman*’ under high-temperature stress during the ripening period. These results suggested that *OsSUT1* contributed to the effective sucrose transport to rice grains, resulting in plant thermotolerance.

## 4. Genes Involved in Antioxidant Defense

One of the major consequences of heat stress is the accumulation of ROS, which leads to oxidative stress. The most common ROS are ^1^O_2_, O_2_^·−^, H_2_O_2_, and OH^·^. Plant oxidative stress tolerance may be improved by increasing the activity of antioxidant enzymes such as superoxide dismutase (SOD), ascorbate peroxidase (APX), and catalase [73,74].

Oxidative stress is associated with a large accumulation of ROS, of which the accumulation of H_2_O_2_ is a typical phenomenon. Hsfs can be involved in heat stress response as directs sensor of H_2_O_2_ in plants [75]. *HsfA* has a transcriptional activation function and is the main regulator of high-temperature-induced gene expression. After heat stress treatment, protoplasts of a *hsfA2* knockout mutant accumulated a much higher concentration of ROS than in WT protoplasts, causing a more rapid decline in the cell viability in the *hsfA2* knockout mutant than in the WT [6]. These results indicate that knockout of *HsfA2* resulted in more severe oxidative stress and more cell death. Clearly, *HsfA2* can protect plants against heat-induced oxidative damage. Amongst these antioxidant enzymes, SOD is an ubiquitous metalloprotein regarded as the first line of defense in plant cells against ROS toxicity. Recombinant thermostable *MnSOD* from *Nerium oleander* showed excellent resistance to temperatures up to 55 °C by stable dimeric and tetrameric states [76]. As an important enzyme for scavenging ROS in chloroplasts, APX plays an important role in high-temperature resistance of plants. A study of the *APX* (*CaAPX*) gene cloned from *Camellia azalea* shows that overexpression of *CaAPX* induces orchestrated reactive oxygen scavenging and enhances heat tolerances in tobacco [35].

## 5. Genes Involved in Heat Signal Transduction

### 5.1. Ca Signaling

Free Ca^2+^ is a universal second messenger, involved in various physiological processes. When cells respond to changes in the external environment, the amount of Ca^2+^ transported from the apoplast through the plasma membrane or intracellular Ca^2+^ stored in the cytosol increases. After the cytoplasmic Ca^2+^ concentration increases, Ca^2+^ sensors transduce Ca^2+^ signals to various substrates, thereby regulating different physiological processes. These plant Ca^2+^ sensors are classified into four major groups: the calmodulin family, the CDPK family, the calcineurin B-like family, and its closely related group, the calmodulin-like protein family [16].

CaM can bind to target proteins and act as part of the Ca^2+^ signal transduction pathway. After plants are exposed to heat stress, the expression of CaM increases, and the concentration of Ca^2+^ increases briefly [77,78]. A heat-induced increase in intracellular Ca^2+^ is one of the earliest cellular changes observed during the plant heat shock response.

LlWRKY39, a member of the WRKYIId transcription factor family of lily (*Lilium* spp.), interacts with LlCaM3 in a Ca^2+^-dependent manner through the CaM-binding domain [15]. The LlWRKY39–LlCaM3 interaction repressed the activation ability of LlWRKY39 toward its target genes. The overexpression of *LlWRKY39* increased the heat tolerance of lily. Further studies revealed that LlWRKY39 can activate the expression of *multiprotein bridging factor 1* (*LlMBF1c*) by directly binding to the *LlMBF1c* promoter. In addition, *MBF1c* is a highly conserved transcriptional co-activator that plays an important role in the heat stress response (HSR) [79]. These results indicated that LlWRKY39 may act as a downstream component of the CaM-mediated Ca^2+^ signaling pathway that lies upstream of *LlMBF1c* in the HSR.

A multi-protein family of CDPKs has been identified as Ca^2+^ sensors in many plant species. CDPKs are directly activated by Ca^2+^ and can transmit Ca^2+^ signals downstream, thus functioning in the regulation of plant growth, development, and abiotic stress responses [80]. ZmCDPK7 is located in the plasma membrane but can translocate to the cytosol under heat stress [7]. *ZmCDPK7*-overexpression maize plants displayed higher thermotolerance, photosynthetic rates, and antioxidant enzyme activity but lower H_2_O_2_ and MDA contents than WT under heat stress. ZmCDPK7 activated the chaperone function of sHSP17.4 via phosphorylation, and positively regulated heat stress tolerance in maize. ZmCDPK7 played a vital role in maintaining protein quality to reduce damage to membranes and the photosynthetic apparatus under heat stress. Under heat stress, expression of the lentil (*L. culinaris*) homologs of *AtCDPKs*, including *AtCDPK4* and *AtCDPK11*, was upregulated. *AtCDPK4* and *AtCDPK11* promoted ethylene biosynthesis, thereby enhancing the heat tolerance of lentils [38].

In addition, cyclic nucleotide-gated ion channels (CNGCs) are non-selective cation channels in the plasma membrane and have been found to play important roles in Ca^2+^ signal transduction [32]. In *Arabidopsis*, AtCNGC6 is a heat- and cAMP-activated plasma membrane Ca^2+^-permeable channel, inducing the production of NO and H_2_O_2_, thereby regulating the expression of *Hsps* and improving the heat tolerance of *Arabidopsis* [23,24,25]. Conversely, *At**CNGC2* deficiency resulted in an increase in the activity of the ROS-quenching enzymes ascorbate peroxidases, Hsps, and MBF1c-dependent heat response pathways in *Arabidopsis* seedlings, indicating its negative role in thermotolerance of *Arabidopsis* at the seedling stage [26,27]. Furthermore, mutants with functional deletions of *OsCNGC14* and *OsCNGC16* reduced tolerance to both heat and cold in rice [32]. These mutants displayed reduced survival rates, higher ROS accumulation, and increased cell death in response to heat stress.

### 5.2. Nitric Oxide Signaling

It has been confirmed that NO acts as a signaling molecule in plant heat stress responses. Endogenous NO levels increase in *Arabidopsis* under heat stress [48]. When wheat is exposed to heat stress, exogenous application of NO enhances the heat tolerance of wheat by reducing H_2_O_2_-induced oxidative stress and photosynthetic suppression [81]. Nitric oxide-associated protein 1 (NOA1) is involved in the regulation of NO [82,83]. Xuan et al. found an increased level of NO in a loss-of-function mutant *noa1* slightly reduced the survival ratio in the condition of heat stress, compared with WT [84]. NO acts as a second messenger for the induction of *AtCaM3* expression under heat stress. It was found that the *AtCaM3* mRNA transcription was strongly inhibited in the *noa1* seedlings under heat stress. Under the care of 20 μM sodium nitroprusside, the *AtCaM3* mRNA transcription levels turned back to the rescued *noa1* line and increased successfully. This result revealed that under heat stress, *AtCaM3* acted as downstream signal transduction of NO. In addition, further research showed that NO regulated the DNA-binding activity of HSFs and the accumulation of HSPs through AtCaM3. Therefore, NO enhanced the thermotolerance of *Arabidopsis*. Peng et al. found that CNGC6, a heat-activated Ca^2+^ permeable channel, could not only mediate Ca^2+^ signaling but also induce NO production [24]. Further research found that CNGC6 regulated internal NO levels by free Ca^2+^ under heat stress. CNGC6 stimulated *HSP* expression with the help of NO. In order to enhance thermotolerance, NO played a role as downstream receptors of CNGC6.

### 5.3. Hormone Signaling

After plants are subjected to biotic or abiotic stress, the production of hormones will help the plants adapt to unfavorable environments [17]. The major hormones produced by plants in response to stresses are abscisic acid (ABA), salicylic acid (SA), ethylene, jasmonate (JA), and gibberellins (GA). Recent studies have provided substantial evidence for ABA, SA, ethylene, and JA in regulating plant heat stress response.

ABA is an important hormone enabling plants to tolerate external stress and mediating stomatal closure to adapt to heat stress. High-temperature stress stimulates ABA signaling in plants and accelerates ABA synthesis [85,86]. A recent study demonstrated that *TaMYB80*, a R2R3-MYB subfamily gene, was involved in ABA-mediated responses to heat stress in wheat [44]. The overexpression of *TaMYB80* in *Arabidopsis* increased ABA levels under heat stress. When exogenous ABA was applied, the expression of *TaMYB80* in wheat seedlings was significantly upregulated. The survival rate of *TaMYB80*-overexpressed seedlings following heat stress was significantly higher than that of the WT, indicating that *TaMYB80* overexpression improved heat tolerance. In addition, under heat stress, compared with WT plants, *AtMYB68*-overexpressed *Arabidopsis* exhibited increased sensitivity to ABA, reduced transpiration, and improved seed yield, showing that *AtMYB68* overexpression improved the heat tolerance of *Arabidopsis* [87]. The overexpression of *AtMYB68* controlled by the heat-inducible promoter P81.1 in *Brassica napus* also improved heat tolerance at the flowering stage, enhancing pollen viability and reducing water loss and transpiration under heat stress. In addition, cell wall-associated protein kinases (WAKs) are typical RLKs [88]. Although it was upregulated under the treatment of ABA, transcription of the WAK-like gene *CaWAKL20* in peppers was downregulated under heat stress [14]. Exposed to high temperatures, the survival rate of *CaWAKL20*-overexpressed *Arabidopsis* plants was decreased. Further research found that ABA pretreatment increased the survival rate of *CaWAKL20*-overexpressed *Arabidopsis* plants under heat stress. Analysis of gene expression under heat stress and ABA pretreatment showed that *CaWAKL20* downregulated the thermotolerance of plants by inhibiting the expression of several ABA-responsive genes. Those ABA-responsive genes encode an abscisic acid-responsive element-binding protein, ABA-responsive element-binding factor, *DREB*, and *HsfA3*.

As an endogenous signal molecule in plants, SA levels can be upregulated under heat stress to alleviate the damage caused to plants by heat stress. Several transcription factors are also known to play important roles in SA signal pathways by mediating defense responses in plants. *Arabidopsis WRKY39* positively regulates SA signal pathways [28]. In response to heat stress, the overexpression of *WRKY39* increased the expression levels of heat-related genes such as *SA-regulated pathogenesis-related 1* and *MBF1c* and enhanced the tolerance to heat stress. *SlJA2*, a transcription factor with a conserved NAC domain, was isolated from tomato [47]. Under heat stress, it was found that the degree of wilting in a *SlJA2*-overexpressed tobacco transgenic line was significantly higher than that of WT. Further studies showed that, in the *SlJA2*-overexpressed tobacco under heat stress, the SA content decreased, stomatal opening increased, the rate of water loss increased, and the plants contained higher concentrations of the ROS H_2_O_2,_ and O_2_^−^. These results showed that the overexpression of *SlJA2* reduced the heat tolerance of tobacco through the SA pathway.

Ethylene is a small gaseous plant hormone molecule. Heat stress can increase the content of ethylene in tomato pollen grains [89]. In the experiment on tomato, ethylene production went up significantly under heat stress (42 °C). The more ethylene produced, the more *HSP70* is expressed [90]. Another study found that pollen grains of tomato plants (cultivar “Hazera 3017”) have the capacity to produce ethylene under heat stress [89]. In mature pollens under heat stress, upregulation of the ethylene receptor *SlETR3* and downstream components of the ethylene-signaling cascade (including *SlCTR2*), and upregulation of several genes involved in ethylene biosynthesis (including *SlACS3* and *SlACS11*) were observed. ERFs play a critical role in ethylene signaling and heat stress response. Results showed that ethylene signaling-defective mutants could weaken basal thermotolerance. Plants with activated ethylene signaling could strengthen basal thermotolerance. Huang et al. reported that two interacting ERFs, ERF95, and ERF97, acted as downstream receptors of ethylene insensitive 3 (EIN3). They were bounded directly to the promoter of *HsfA2*. This showed a EIN3-ERF95/ERF97-HSFA2 transcriptional cascade under heat stress [91]. In addition, 1-Methylcyclopropene, an ethylene resistance agent could weaken the physiological effect of ethylene by preferentially attaching ethylene receptors. Analysis of Chinese bayberry (*Myrica rubra Sieb. et Zucc.*) fruit transcriptome from postharvest storage revealed that most ethylene receptors (Unigene21949_All, Unigene3820_All, Unigene8475_All, Unigene13442_All, Unigene23855_All, and Unigene23780_All) and ERFs (Unigene6615_All, Unigene12612_All, Unigene21144_All and Unigene24054_All) were downregulated because of heat stress and 1-MCP treatment [50]. At the same time, ABF (CL3405.Contig1_All), PR-1C (Unigene1312_All), GH3 (Unigene13051_All), and MYC2 (Unigene23741_All) related to ethylene signaling were also downregulated.

JA and its derivatives including methyl jasmonic acid (MeJA) together are called JA. JA is necessary for the activation of defense response against necrotrophic pathogens [92]. JA positively regulates thermotolerance in *Arabidopsis* by physiological protection from heat-induced damage, and the application of exogenous JA also can increase thermotolerance [93]. The study showed that MeJA could significantly improve the heat tolerance of perennial ryegrass (*Lolium perenne* L.) through the alteration of osmotic adjustment, antioxidant defense, and the expression of JA-responsive genes [94]. Heat stress and exogenous MeJA upregulated transcript levels of related genes (*LpLOX2*, *LpAOC*, *LpOPR3*, and *LpJMT*) and *LpHsp70* in JA biosynthetic pathway, which enhanced the accumulation of JA and MeJA content. Yu et al. found that JA levels were higher in thermotolerant cucumber than in thermosensitive cucumber [95]. Heat stress specifically enriched the lipid metabolism, metabolism of amino acids, and biosynthesis of secondary metabolites in thermosensitive cucumber. Repair, catabolism, and energy metabolism were specifically enriched in thermotolerant cucumber.

## 6. Other Regulatory Genes

NcRNAs are a type of RNA that has no obvious open reading frame and can directly perform biological functions without coding for a protein. NcRNAs can be classified into small RNAs (18–30 nucleotides [nt]), medium-sized ncRNAs (31–200 nt), and lncRNAs (>200 nt) [96].

### 6.1. MiRNAs

MiRNAs are a major class of small RNAs in plants, with a length of 18–36 nt. They degrade or inhibit the translation process of their target genes through sequence complementation, thereby regulating gene expression at the post-transcriptional level [97,98]. MiR156 subtypes have been reported to be induced by heat stress. After four-day-old *Arabidopsis* seedlings were exposed to heat stress, the expression of *miR156* was highly upregulated. By inhibiting the expression of SQUAMOSA promoter-binding protein-likes, miR156 maintained the expression of heat stress response genes (such as *HsfA2* and *Hsps*), thereby increasing the expression of *Arabidopsis* thermotolerance [18]. MiR159 is involved in the control of GA signal transduction and *GAMYB* transcription factors that negatively regulate pollen development [99]. After wheat seedlings were subjected to heat stress, the transcription of *miR159* was downregulated, leading to the accumulation of the mRNA of its target gene *TaGAMYB*, which participates in heat stress response [51,100]. Studies found that *miR159*-overexpressed rice showed increased heat sensitivity. When two-week-old sunflower (*Helianthus annuus*) seedlings were subjected to heat stress, the transcription of *miR396* was downregulated, which increased the expression of the WRKY family transcription factor *HaWRKY6*, thereby helping the sunflower to be protected against heat stress damage [52]. Studies found that transgenic sunflowers expressing a *miR396*-resistant form of *HaWRKY6* were more sensitive to heat stress. After fifteen-day-old *Arabidopsis* was subjected to heat stress, the expression of *miR398* was upregulated, which downregulated the transcription of its target genes copper/zinc superoxide dismutase 1 (*CSD1*), *CSD2*, and copper chaperone for SOD [101]. This reduced the accumulation of ROS, which enhanced the heat tolerance of *Arabidopsis*. Studies found that transgenic *Arabidopsis* expressing the *miR398*-resistant forms of *CSD1*, *CSD2*, or copper chaperone for SOD were more sensitive to heat stress. In the heat-stressed banana (*Musa* spp. AAA group, cv. Cavendish), *miR164* and *miR168* were accumulated. Both *miR159* and *miR396* were greatly repressed by heat stress [49].

### 6.2. LncRNAs

LncRNAs are a type of non-coding RNA that is widely present in eukaryotes, with a length greater than 200 nt and extremely weak transcription ability. LncRNAs are involved in plant growth and development, nutrient metabolism, biotic and abiotic stress responses, and other biological processes [102].

Song et al. used strand-specific RNA sequencing to identify 204 high-temperature-responsive lncRNAs in poplar (*Populus simonii*) [19]. In response to heat stress, the expression of *TCONS_00202587* and *TCONS_00260893* were significantly upregulated. Compared to the control group under high temperatures, silencing of *lncRNA TCONS_00202587* and *TCONS_00260893* resulted in the significant downregulation of their target Potri.017G089800 (CNGC2) and Potri.012G002800 (protein phosphatase 2C), respectively. Further experiments showed that high-temperature-responsive TCONS_00202587 and TCONS_00260893, regulated their targets via RNA interference or acted as RNA scaffolds, thereby promoting photosynthetic protection and recovery under heat stress. Wang et al. analyzed the lncRNAs of three-week-old Chinese cabbage seedlings and verified three differentially expressed lncRNAs (*TCONS_00017642*, *TCONS_00053114*, and *TCONS_00004594*) in response to heat stress by strand-specific RNA-sequencing [103]. Among them, *lncRNA TCONS_00004594* cis-regulated the expression of *Bra021232* downstream of the protein-coding gene. Heat treatment of the heat-tolerant Chinese cabbage variety ‘XK’ found that the expression of *bra-miR164a* was upregulated, whereas the expression of eTM (*TCONS_00048391*) and its target genes (*Bra030820* and *NAC1*) were downregulated. The expression levels of lncRNAs were estimated by the fragments per kilobase per million fragments mapped value, using Cuffdiff. Compared with the controlled group, 108 lncRNAs in three-leaf-stage cucumber (*Cucumis sativus* L.) seedlings leaves expressed differentially under heat stress. 56 lncRNAs were upregulated while 52 lncRNAs were downregulated [20]. The study found that the protein-encoding genes *Csa4M314390.1* (*Erf*) and *Csa5M613470.1* (*Myb_Related*) were involved in the response. The competing endogenous RNA network indicated that TCNS_00031790, TCNS_00014332, TCNS_00014717 and TCNS_00005674, NOVE_CIRC_001543, and NEWE_CIRC_000876 might interact with miR9748 and regulate heat shock response through miR9748 and its target genes in the plant hormone signaling pathway.

## 7. Summary and Prospects

Under the current climate of global warming, high-temperature stress has aroused great concern. Plants have evolved complex and efficient mechanisms of sensing and responding to high temperatures, consisting of the activation of numerous regulatory and signaling pathways that eventually lead to a fine metabolic adjustment to achieve survival (Figure 1) [104]. Among these mechanisms, Hsf and Hsp play key roles in HSR. In addition, we also found that protecting the normal synthesis of chlorophyll and reducing the production of ROS play important roles in enhancing plant heat tolerance. The signaling pathway includes Ca^2+^, NO, and hormone signaling. They could help plants adapt to high-temperature environments.

Due to expressions of genes, plants at different growing stages respond to heat stress differently. For example, *Arabidopsis* at the stage of germination requires the expression of *CaHsfA1d*, *OsHIRP1*, and *AtPUB48* under heat stress. *Arabidopsis* at the stage of seedling requires the overexpression of *ZmHSF01*, *OsHsp20*, *TaMYB80*, *miR156*, etc. *Brassica napus* at the flowering stage requires the overexpression of *AtMYB68* under heat stress. Tomato pollens require the expressions of *SlACS3* and *SlACS11* under heat stress. Rice grains between photoassimilates and filling grains require the expression of *OsSUT1* under heat stress.

The response of plants to heat stress is a very complex process. It is worth noting that many published studies only show the expression of heat-regulated genes in response to heat stress, but do not detail the upstream regulatory mechanisms and downstream target genes. Future research should focus mainly on the upstream mechanisms, such as heat stress sensing, which activate transcriptional cascades. Functional genomics, proteomics, and transcriptomics are required to research the response of plants to high-temperature stress. For example, the third-generation gene editing technology, namely CRISPR-Cas9, will be a powerful tool in this endeavor [105]. Furthermore, recently developed technologies, e.g., assay for transposase-accessible chromatin sequencing, RNA sequencing, chromatin immunoprecipitation sequencing, RNA modification (m6A/m1A/m5C), and single-cell RNA sequencing will be essential to elucidate gene networks involved in plant heat stress response and tolerance.

## Figures and Tables

**Figure 1 ijms-23-11970-f001:**
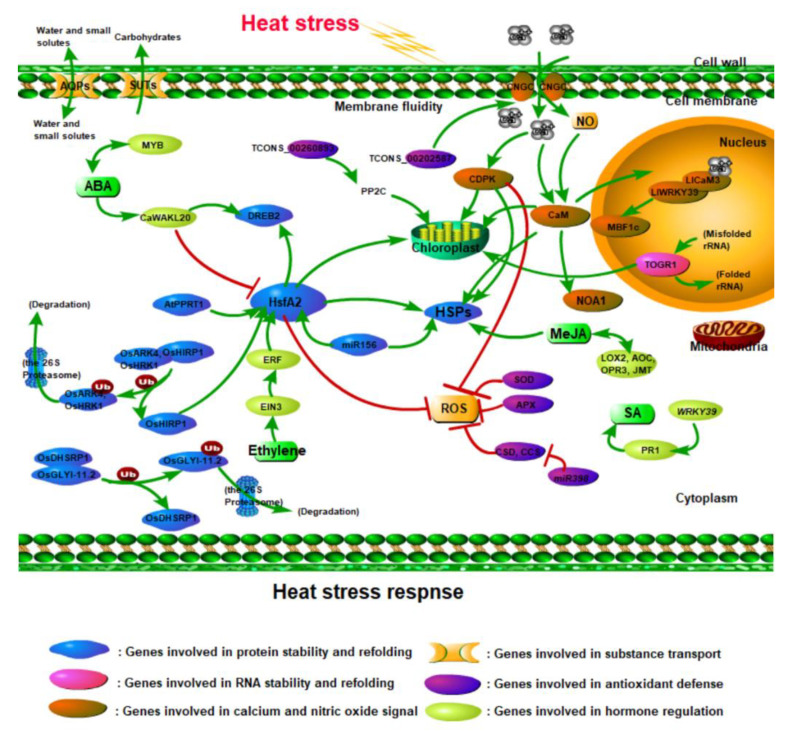
Gene networks in plant heat stress response. The large regulatory networks involve genes related to protein and RNA stability and refolding, Ca^2+^, NO, and hormone signaling, substance transport, and antioxidant defense. Heat stress alters membrane fluidity which may activate Ca^2+^ channels resulting in an influx of Ca^2+^. Ca^2+^ signals are transduced by CaM (calmodulin) and CDPK (calcium-dependent protein kinase), activating signal transduction pathways in plants in response to heat stress. NO regulates the accumulation of HSPs (heat shock proteins) through AtCaM3. LlWRKY39 interacts with LlCaM3 in a Ca^2+^-dependent manner through the CaM-binding domain, and it promotes the expression of LlMBF1c. Under heat stress, HsfA2s are activated by *AtPPRT1*, *OsHIRP1*, *ERF*, and *miR156*. The expression of *HsfA2* is activated to upregulate or fine-tune the expression of *DREB2* (dehydration response element binding protein 2) and *HSP**s*. As the main functional proteins induced by heat stress, HSPs and ROS (reactive oxygen species) constitute a complex regulatory network with CDPK, HsfA2, and some antioxidant enzymes. Both OsHIRP1 and OsDHSRP1 are degraded by the Ub/26S proteasome system to participate in the HSR (heat stress response) process. Under heat stress, *TOGR1* expression is enhanced in the nucleolus, which helps the rRNA precursor to effectively fold. TOGR1, CDPK, and TCONS_00260893 protect chlorophyll synthesis under heat stress. AQPs (aquaporins) and SUTs (sucrose transporters) play important roles in maintaining the normal transport of water and sucrose under heat stress. In addition, some hormone-related genes are upregulated to promote hormone synthesis, such as *MYB*, *EIN3*, *LOX2*, *AOC*, *OPR3*, *JMT*, and *WRKY39*. Arrows denote the positive while red bars stand for negative interaction.

**Figure 2 ijms-23-11970-f002:**
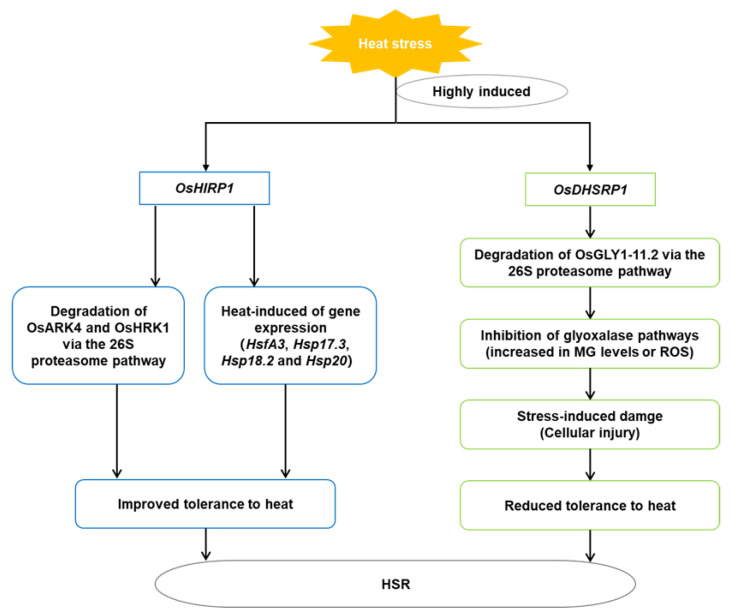
Schematic diagram of expression of *OsHTRP1* and *OsDHSRP1* under high-temperature stress in rice. Under heat stress, the expression of *OsHIRP1* and *OsDHSRP1* increases significantly. *OsARK4* and *OsHRK1* are ubiquitinated by OsHIRP1 E3 ligase, and each is degraded by the Ub/26S proteasome system. OsHIRP1 promotes the expression of some heat stress-inducible genes (*HsfA3*, *Hsp17.3*, *Hsp18.2*, and *Hsp20*) under heat stress. OsDHSRP1 E3 ligase ubiquitinates OsGLYI-11.2, which is degraded by the Ub/26S proteasome system. The degradation of the OsGLYI-11.2 protein maintains a lower acetaldehyde level, thus increasing methylglyoxal and ROS contents in *Arabidopsis* plants overexpressing *OsDHSRP1*.

**Table 1 ijms-23-11970-t001:** Upregulated genes involved in plant heat stress response.

Plant	Gene	Protein	Function	Reference
*Arabidopsis thaliala*	*AtPUB48*	Ubiquitin E3 ligase	Protein stability and refolding	[22]
	*AtCNGC2*, *AtCNGC6*	CNGC (cyclic nucleotide gated ion channel)	Ca^2+^ signal transduction	[23,24,25,26,27]
	*WRKY39*	WRKY39	Salicylic acid (SA) signal regulation	[28]
	*miR156*	/	Protein stability and refolding	[18]
Rice (*Oryza sativa*)	*OsHSP18.0*	HSP (heat shock protein)	Protein stability and refolding	[29]
	*OsHIRP1*	Heat-induced RING finger protein 1	Protein stability and refolding	[10]
	*OsDHSRP1*	Drought, Heat and Salt-induced RING finger protein 1	Protein stability and refolding	[30]
	*eIF4A1*	DEAD-box RNA helicase	RNA stability and refolding	[31]
	*TOGR1*	DEAD-box RNA helicase	RNA stability and refolding	[11]
	*CNGC14*, *CNGC16*	CNGC	Ca^2+^ signal transduction	[32]
Apple (*Malus domestica*)	*MdATG18*	Autophagy-related proteins	Photosynthesis	[33]
Cabbage (*Brassica rapa*)	*BrRH22*	DEAD-box RNA helicase	RNA stability and refolding	[34]
*Camellia azalea*	*CaAPX*	APX (ascorbate peroxidase)	Antioxidant defense	[35]
Grape (*Vitis vinifera*)	*HSFA6B*	HSF	Protein stability and refolding	[36]
Jujube (*Ziziphus jujuba* Mill.)	*HSP17*, *HSP18*, *HSP21*, *HSP 22*, *HSP 23*, *HSP26*, *HSP70*, *HSP83*, *HSP90*, *HSF30*, and *HSC-2*	HSP	Protein stability and refolding	[37]
Lentil (*L. culinaris*)	*AtCDPK4 AtCDPK11*	CDPK (Calcium-dependent protein kinase)	Ca^2+^ signal transduction	[38]
Lily (*Lilium* spp.)	*LlWRKY39*	WRKY39	Ca^2+^ signal transduction	[15]
Maize (*Zea mays* L.)	*ZmHsf01*	HSF	Protein stability and refolding	[39]
	*ZmCDPK7*	CDPK	Ca^2+^ signal transduction	[7]
Pepper (*Capsicum annuum* L.)	*CaHsfA1d*	HSF (heat shock factor)	Protein stability and refolding	[40]
poplar (*Populus simonii*)	*TCONS_00202587*, *TCONS_00260893*	/	Antioxidant defense	[19]
*Rhazya stricta*	*Pip2-1*, *pip1-2*, *tip21*	AQP	Substance transport	[41]
Sorghum (*Sorghum bicolor*)	*SbHCI1*	Heat- and cold-induced RING finger protein 1	Protein stability and refolding	[42]
Wheat (*Triticum aestivum*)	*TaFBA1*	Ubiquitin E3 ligase	Protein stability and refolding	[43]
	*TaMYB80*	MYB	Abscisic acid (ABA) signal regulation	[44]
Tomato (*Solanum lycopersicum*)	*HsfB1*	HSF	Protein stability and refolding	[45]
	*SlDEAD31*	DEAD-box RNA helicase	RNA stability and refolding	[46]
	*SlJA2*	*Solanum lycopersicum* jasmonic acid 2	Salicylic acid (SA) signal regulation	[47]

**Table 2 ijms-23-11970-t002:** Downregulated genes involved in plant heat stress response.

Plant	Gene	Protein	Function	Reference
Rice	*OsSUT1*	Rice sucrose transport protein	Substance transport	[48]
Banana (*Musa acuminata*)	*miR159*	/	Hormone regulation	[49]
	*miR396*	/	Leaf development	[49]
Chinese bayberry (*Myrica rubra Sieb. et Zucc.*)	*Unigene21949_All*, *Unigene3820_All*, *Unigene8475_All*, *Unigene13442_All*, *Unigene23855_All*, and *Unigene23780_All*	Ethylene receptor	Ethylene signal regulation	[50]
	*Unigene6615_All*, *Unigene12612_All*, *Unigene21144_All*, and *Unigene24054_All*	Ethylene response factors (ERFs)	Ethylene signal regulation	[50]
Pepper	*CaWAKL20*	WAK (cell wall-associated protein kinase)	ABA signal regulation	[14]
Wheat	*miR159*	/	Hormone regulation	[51]
Sunflower (*Helianthus annuus*)	*miR396*	/	Leaf development	[52]
Tomato	*SlDEAD30*	DEAD-box RNA helicase	RNA stability and refolding	[46]

## Data Availability

Data sharing is not applicable to this article as no new data were created or analyzed in this study.

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
