# Peer review of "Gene Networks Involved in Plant Heat Stress Response and Tolerance"

_ijms, 2022, doi:10.3390/ijms231911970_

Round 1

Reviewer 1 Report

The review paper is quit good and comprehensive for almost works on plant heat shock genes. It is also huge work and complicated one's that we recommend the author to do double check for obtained information from various references that I don,t doubt he/they did .

Author Response

请参阅附件。

Reviewer 2 Report

Dear Authros,

The submitted manuscript is of great importance in the field of plant stress genetics, as global climate change is increasing the frequency of extreme weather events, including heat stress.

It is therefore important that work of this kind should cover this area adequately. 

In my opinion, the basic concept of the manuscript is good, but it needs to be modified both in its scope and structure and in the number of references included.

Firstly, plant heat stress is such a broad field that a book rather than a journal article would be appropriate for a general plant article. Firstly, since different plants, depending on their habitat, whether they are monocotyledonous or dicotyledonous, wild or cultivated, all respond differently to heat stress, in addition to the general mechanisms.

Therefore, I consider the title and the topic in its current form to be too broad an approach, and therefore suggest narrowing the topic or incorporating sub-chapters in a structured way, using more literature. This will produce a work that will be of real benefit to the scientific community.

Reviewer 3 Report

In the review article by Huang et al. entitled “Gene networks involved in plant heat stress response and tolerance”, the authors have summarized recent research results on the molecular mechanisms of plant response to high temperature emphasizing future strategies for understanding the plant’s response to heat stress. While the authors have discussed important mechanisms and networks, the gene information seems limited, especially when a large number of genes have been reported across the plants.

The basis behind the gene list is unclear, as there are a plethora of genes identified for HS response in plants. An account of such genes can be given in the supplementary information. Also, many of these genes have been upregulated, and the authors have mentioned in the conclusion that fewer studies are reporting downregulated genes. A list of such genes and their roles can be given and discussed. Several knock-out studies of genes such as MADP have been reported. The authors can include such genes, which will provide a better understanding of gene networks. Also, several studies in perennial horticultural crops have reported downregulated genes in the last few years. The authors are suggested to include those responses. 

The genes (especially the nuclear heat-response genes) related to photosynthetic processes including chlorophyll biosynthesis, photochemical reactions, electron transport, and CO2 assimilation, are given less attention but are necessary for a complete understanding of the gene networks involved in HS response. 

The responses by the plants to heat stress, differ based on the stage of the plants, and whether the plants were subjected to moderate or severe temperature stress. Thus, stage-wise differences need to be given attention, as it will help develop strategies for mitigating the temperature effects. 

The authors did not discuss the gene networks related to nitric oxide signaling in plant heat stress. Also, many genes have been reported to be involved in ethylene response to heat stress; however, the authors have discussed only a single paper. Thus, the authors are suggested to include recent related studies at least in tabular formats. 

Some minor suggestions

  1. The botanical names of plants need to be given at the first instance of appearances in the article. 
  2. High-resolution Figure 1 needs to be given
  3. The abbreviation of ethylene can be avoided
  4. Large errors in the reference list as the suggested format is not followed properly. 

Round 2

Reviewer 3 Report

It is suggested to re-write the modified sections for further clarity and flow
